# Hormone Regulation of CCCH Zinc Finger Proteins in Plants

**DOI:** 10.3390/ijms232214288

**Published:** 2022-11-18

**Authors:** Qiao Wang, Shangfa Song, Xintong Lu, Yiqing Wang, Yan Chen, Xiuwen Wu, Li Tan, Guohua Chai

**Affiliations:** 1College of Resources and Environment, Qingdao Agricultural University, Qingdao 266109, China; 2College of Landscape Architecture and Forestry, Qingdao Agricultural University, Qingdao 266109, China; 3Complex Carbohydrate Research Center, University of Georgia, Athens, GA 30602, USA; 4Institute of Efficient Agricultural Carbon Neutrality in Middle-Lower Yellow River Regions, Qingdao 266109, China

**Keywords:** CCCH zinc finger protein, development and stress, hormone response, multifaceted regulation

## Abstract

CCCH zinc finger proteins contain one to six tandem CCCH motifs composed of three cysteine and one histidine residues and have been widely found in eukaryotes. Plant CCCH proteins control a wide range of developmental and adaptive processes through DNA–protein, RNA–protein and/or protein–protein interactions. The complex networks underlying these processes regulated by plant CCCH proteins are often involved in phytohormones as signal molecules. In this review, we described the evolution of CCCH proteins from green algae to vascular plants and summarized the functions of plant CCCH proteins that are influenced by six major hormones, including abscisic acid, gibberellic acid, brassinosteroid, jasmonate, ethylene and auxin. We further compared the regulatory mechanisms of plant and animal CCCH proteins via hormone signaling. Among them, Arabidopsis AtC3H14, 15 and human hTTP, three typical CCCH proteins, are able to integrate multiple hormones to participate in various biological processes.

## 1. Introduction

Zinc finger proteins (ZFPs) are ubiquitous in eukaryotes with diversified structural features of zinc finger domains. Based on the numbers and orders of conserved cysteine (C) and histone (H) residues, ZFPs are classified into nine distinct types, including C_2_H_2_, C_3_H, C_2_HC, C_3_HC_4_, C_2_HC_5_, C_4_HC_3_, C_4_, C_6_ and C_8_. Members of these ZFP families play critical roles in cell differentiation, proliferation and apoptosis through the transcriptional and/or post-transcriptional regulations of downstream gene expression [1,2,3].

The CCCH−type ZFPs contain one to six tandem CCCH motifs and have been identified in many organisms, such as human (*Homo sapiens*, 55 members) [4], mouse (*Mus musculus*, 58 members) [4], Arabidopsis (*Arabidopsis thaliana*, 68 members) [5], rice (*Oryza sativa*, 67 members) [5], poplar (*Populus trichocarpa*, 91 members) [6], moso bamboo (*Phyllostachys edulis*, 119 members) [7] and malaria (*Plasmodium falciparum*, 27 members) [8]. Currently, the biological functions and regulatory mechanisms of animal CCCH proteins have been documented in detail, compared with plants. The best-studied CCCH protein is human Tristetraprolin (hTTP)/ZFP36, which contains two identical, tandem C-X8-C-X5-C-X3-H motifs separated by 18 amino acids, belonging to typical tandem CCCH Zinc Finger proteins (TZFs) [9]. hTTP/ZFP36 is able to be rapidly activated by serum, insulin and cytokinin and regulates multiple biological processes, such as metabolic disease, inflammatory reaction and embryo development [10]. At the molecular level, hTTP functions through the protein–DNA, protein–RNA and protein–protein interactions. Similar to animal CCCHs, plant CCCH proteins also regulate growth, development and stress responses by a multi–leveled regulatory mechanism, which has been summarized in three previous reviews [11,12,13]. Among the TZF proteins, members containing a unique TZF domain preceded by an arginine–rich motif are referred to as RR-TZF proteins. The RR-TZF proteins are mainly involved in hormone– and environmental cues–mediated plant growth and stress responses [12]. For example, Arabidopsis AtTZF1 acts as both a positive regulator of abscisic acid (ABA)/sugar responses and negative regulator of gibberellic acid (GA) responses [14].

In this review, we identified the CCCH family members of representative plants, observed their evolutionary relationship, and discussed how plant CCCH proteins affect hormone homeostasis and signaling to control developmental processes and stress responses. We further compared the complicated regulatory mechanisms of CCCH proteins between plants and animals.

## 2. Evolutionary Conservation of the CCCH Proteins in Plants

The phylogenetic analysis of gene family members from different species can give us important clues about the underlying evolutionary process. Here, a total of 342 CCCH proteins were identified from six representative plant species, including two primitive plants (green algae, *Chlamydomonas reinhardtii*, and moss, *Physcomitrium patens*), three angiosperms (Arabidopsis, *Arabidopsis thaliana*; rice, *Oryza sativa*; and poplar, *Populus trichocarpa*) and one gymnosperm (spruce, *Picea abies*). Based on the alignment of the full–length protein sequences of these CCCHs, a Neighbor−Joining (NJ) phylogenetic tree was generated. Bootstrap analysis with 1000 replicates was conducted for statistical reliability [15]. As shown in Figure 1, the 342 CCCH proteins were generally divided into 10 subfamilies. In most subfamilies, the members exhibited an alternating distribution of six detected species, similar to the distribution patterns of CCCH proteins in each subfamily, as we previously observed in Arabidopsis, rice and poplar [6]. These results suggest an evolutionary conservation of CCCH proteins from primitive plants to vascular plants. Notably, most CCCH proteins from the green algae clustered in subfamily I, implying functional divergence of CCCH proteins between green algae and land plants.

The CCCH proteins in subfamily VIII or IX are likely involved in hormone responses. For instance, in subfamily VIII the RR-TZF proteins AtTZF1/AtC3H23, AtTZF2/AtC3H20, AtTZF3/AtC3H49, AtTZF4/AtC3H2, AtTZF5/AtC3H61, AtTZF6/AtC3H54, OsDOS/OsC3H2 and OsTZF1/OsC3H35 have been shown to be responsive to ABA or jasmonate (JA) [14,16,17,18,19,20,21,22,23]. AtC3H14 and 15 and PtC3H17 and 18 in subfamily IX regulate various biological processes through the coordination of multiple hormone signaling pathways [24,25,26,27,28,29,30,31]. These four CCCH proteins are the orthologs of hTTP, all containing two tandem C-X_8_-C-X_5_-C-X_3_-H motifs separated by 18 amino acids and conserved lead-in sequences (MM/TKTEL or RYKTEV) at their N−termini. It will be interesting to investigate which hormone(s) induce(s) or repress(es) the expression of CCCHs in subfamily VIII and IX.

## 3. Responses of Plant CCCH Proteins to Several Major Hormones

Extensive studies in Arabidopsis and rice show that most CCCH proteins regulate plant growth, development and adaptative responses via mediating hormones, such as ABA, GA, JA, ethylene (ET), brassinosteroids (BRs), salicylic acid (SA) and auxin. In this review, we summarized the characteristics of plant CCCH proteins with known functions and their responses to these hormones in Table 1.

### 3.1. ABA Signal Response

ABA is an important hormone in plants, whose signaling pathway runs from the membrane localized PYRABACTIN RESISTANCE 1 (PYR1) receptor to a series of ABA RESPONSIVE ELEMENT-BINDING FACTORs (ABFs), thereby regulating the expression of ABA–responsive genes in the nucleus [32]. In Arabidopsis, *AtTZF1*–*6* act as key regulators of ABA-mediated seed germination and stress responses [14,16,17,18,19,20]. *AtTZF1* is strongly induced by exogenous ABA. In the presence of ABA, *AtTZF1* RNAi plants germinated faster than wild-type plants, while *AtTZF1* overexpression plants displayed an opposite phenotype [14]. AtTZF4/SOMNUS is a component of the phytochrome signal transduction pathway [20]. It positively regulates ABA metabolic genes downstream of PHYTOCHROME-INTERACTING FACTOR3-LIKE5 (PIL5), a phytochrome–interacting basic helix-loop-helix transcription factor, during seed germination. AtTZF2/OZF1 and AtTZF3/OZF2 enhance plant tolerance toward drought, oxidative and salt stresses by antagonizing ABA INSENSITIVE 2 (ABI2), a member of the negative regulators PROTEIN PHOSPHATASE 2Cs (PP2Cs) in the ABA pathway [17,18]. In addition, *ZINC K-HOMOLOG 1*(*AtKHZ1*, *2*) and *AtKHZ2* redundantly and positively regulate ABA−induced leaf senescence in Arabidopsis [33].

Several studies in other species also confirm that plant CCCH genes mediate ABA biosynthesis and signaling to control developmental and adaptive processes (Table 1). For instance, rice RR-TZF genes *OsTZF1*, *OsTZF7*, *OsC3H10* and *OsC3H47* are significantly induced by ABA application [24,34,35,36]. OsTZF1 inhibits ABA–mediated seed germination and seedling growth by altering the RNA metabolism of stress–responsive genes [23]. OsTZF7 positively regulates drought response via ABA signaling and shows potential involvement in mRNA turnover [34]. OsC3H10 and OsC3H47 positively regulate the drought tolerance pathway by modulating the expression of stress–related genes and are involved in ABA feedback [35,36]. The functional study of bamboo *PeC3H74* in Arabidopsis showed that this gene might induce stomatal closure through ABA signaling to achieve drought–resistance [7]. Switchgrass *PvC3H69* is a repressor for ABA-mediated leaf senescence [37]. The ectopic expression of *PvC3H69* in rice suppressed the expression levels of genes involved in ABA biosynthesis (*NCED3*, *NCED5* and *AAO3*) and ABA signaling (*SnRKs*, *ABI5* and *ABF2/3/4*), resulting in delayed leaf senescence. In sweet potato, the up-regulation and down-regulation of *IbC3H18* expression change ABA contents and affect adaptation to short-term and long-term salt, drought and MV treatments [38].

### 3.2. GA Signal Response

GAs are a class of diterpenoid phytohormones that modulate diverse processes throughout plant growth and development [39]. Arabidopsis AtTZF1, 4, 5 and 6 function as negative regulators for GA accumulation and response, in addition to serving as positive regulators for ABA signaling (Table 1). Resembling to GA-deficient mutants, transgenic plants overexpressing *AtTZF4*, *5* or *6* displayed compact and dark green rosettes and late flowering phenotypes accompanied by the lower levels of GA, compared with wild-type plants [19]. Furthermore, the early germination phenotype of the mutants was repressed by exogenous ABA application, while the late germination phenotype of the overexpression plants was rescued by addition of exogenous GA. These findings suggest that AtTZF4, 5 and 6 might regulate seed germination through the coordination of ABA and GA.

In rice, the CCCH protein SWOLLEN ANTHER WALL 1 (SAW1) positively regulates the development of anther walls through the GA signaling pathway [40]. Loss-of-function mutant *saw1* showed a lower content of GA and displayed disordered and enlarged anther wall layers and disrupted structures of pollen exine and intine. The fertile pollen rate of *saw1* was partially restored by the application of exogenous GA. Consistent with this, SAW1 activated *OsGA20ox3* expression, which led to an increase of GA12 level during GA homeostasis.

**Table 1 ijms-23-14288-t001:** Characteristics of CCCH proteins with known functions and their responses to different hormones.

Species	Corresponding Gene	ID	NO. of CCCH Motif	Subcellular Location	Responsive Hormone	Reference(s)
	*AtTZF1/AtC3H23*	AT2G25900	2	Nucleus/cytoplasm	ABA/GA	[14]
	*AtTZF2/OZF1/AtC3H20*	AT2G19810	2	Cytoplasm	ABA/SA/JA	[16,17]
	*AtTZF3/OZF2/AtC3H49*	AT4G29190	2	Cytoplasm	ABA/JA	[16,18]
	*AtTZF4/AtC3H2/SOMNUS*	AT1G03790	2	Nucleus	ABA/GA	[19,20]
	*AtTZF5/AtC3H61*	AT5G44260	2	Cytoplasm	ABA/GA	[19]
	*AtTZF6/PEI1/AtC3H54*	AT5G07500	2	Cytoplasm	ABA/GA	[19,21]
	*AtKHZ1/AtC3H36*	AT3G12130	2	Nucleus	ABA	[33]
	*AtKHZ2/AtC3H52*	AT5G06770	2	Nucleus	ABA	[33]
	*AtC3H14*	AT1G66810	2	Nucleus/cytoplasm	JA/ET/SA/BR	[24,25,26,27,28,29]
	*AtC3H15*	AT1G68200	2	Nucleus/cytoplasm	JA/ET/SA/BR	[24,25,26,27,28,29]
	*AtCPSF30/AtC3H11*	AT1G30460	3	Nucleus	Auxin/SA	[41,42]
	*AtHIZ1*	AT1G32360	3	Nucleus	Auxin	[43]
rice	*OsTZF1/OsC3H35*	LOC_Os05g10670	2	Cytoplasm	ABA/JA/SA	[23]
	*OsTZF7*	LOC_Os10g37630	2	Nucleus/cytoplasm	ABA	[34]
	*OsC3H10*	LOC_Os01g53650	2	Nucleus/cytoplasm	ABA	[35]
	*OsC3H47*	LOC_Os07g04580	2	-	ABA	[36]
	*OsSAW1*	LOC_Os06g43120	5	Nucleus	GA	[40]
	*OsDOS/OsC3H2*	LOC_Os01g09620	2	Nucleus	JA	[22]
	*OsC3H12*	LOC_Os01g68860	5	Nucleus/cytoplasm	JA	[44]
	*OsLIC/OsC3H46*	LOC_Os06g49080	1	Nucleus/cytoplasm	BR	[45,46]
poplar	*PaC3H17*	Potri.004G095100	2	Nucleus/cytoplasm	Auxin/BR	[34,35]
	*PaC3H18*	Potri.017G119900	2	Nucleus/cytoplasm	Auxin/BR	[34,35]
bamboo	*PeC3H74*	PH02Gene33725	2	Cytoplasm	ABA	[7]
switchgrass	*PvCCCH69*	Pavir.J04795.1	2	Nucleus	ABA	[37]
sweet potato	*IbC3H18*	MK396199	1	Nucleus	ABA	[38]
cotton	*GhZFP1*	EF403655	3	-	JA	[47]
pepper	*CaC3H14*	CA10g20930	3	-	JA/ET/SA	[48]

Note: ‘-’ means no data in the references.

### 3.3. JA/ET Signal Response

JA and ET coordinately control diverse aspects of plant growth, development and immunity [49,50]. Arabidopsis AtTZF2 and 3 are involved in the negative regulation of JA–induced senescence (Table 1). The overexpression of *AtTZF2* or *AtTZF3* caused a significant alteration in the expression levels of several JA-associated genes and delayed JA-induced senescence [16]. The ectopic expression of cotton *GhZFP1* in Arabidopsis also delayed JA-induced leaf senescence in darkness and enhanced drought tolerance [47]. Arabidopsis AtC3H14 and AtC3H15 are the homologs of hTTP, and they mediate multiple hormone signaling pathways to regulate various biological processes (Figure 2). In the JA and ET signaling pathways, *OCTADECANOID-RESPONSIVE FACTOR59* (*ORA59*), an essential integrator of JA and ET signaling, was identified as a positive target gene of AtC3H14 [26]. AtC3H14 enhanced plant resistance against *Botrytis cinerea* partially by the ORA59-dependent JA/ET signaling. OsDOS is an AtTZF1 homolog in the rice TZF subfamily I, and it delays leaf senescence by integrating developmental cues to the JA pathway [22]. The *OsDOS* RNAi lines showed the phenotypes of enhanced dark-induced leaf senescence, shorter in plant height and slightly earlier flowering time, compared with wild-type plants. When treated with MeJA in darkness, the leaves of its *OsDOS* RNAi plants exhibited a severe senescence symptom, but JA-induced senescence was delayed in the leaves of overexpression lines. Rice OsC3H12 is another JA-responsive CCCH protein. It positively and quantitatively regulates resistance against the rice bacterial blight disease by JA accumulation and induced the expression of JA signaling genes [44].

### 3.4. BR Signal Response

BRs are poly-hydroxylated steroidal hormones that are essential for plant growth through the promotion of cell elongation and differentiation [51]. The molecular components of canonical BR signaling pathways, including the plasma membrane-anchored co-receptor complex (BRASSINOSTEROID INSENSITIVE 1, BRI1 and BRI1-ASSOCIATED RECEPTOR KINASE 1, BAK1), the cytoplasmic and nuclear signal transmitters (BR SIGNAL KINASE 1, BSK1; CONSTITUTIVE DIFFERENTIAL GROWTH1, CDG1; BRI1-SUPPRESSOR1, BSU1; 14-3-3 proteins; BR-INSENSITIVE 2, BIN2 and PROTEIN PHOSPHATASE 2, PP2A) and the master transcription factors (BRASSINAZOLE-RESISTANT 1, BZR1 and BRI1-EMS-SUPPRESSOR 1, BES1), have been well established in plants [52,53]. Recently, we found that Arabidopsis C3H14 and 15-mediated BR signaling is parallel to, or even attenuates, the dominant BZR1 and BES1 signaling [29] (Figure 2). In the absence of BRs, C3H15 is phosphorylated by BIN2 at Ser111 in the cytoplasm. Upon BR perception, *C3H15* transcription is enhanced and the phosphorylation of C3H15 by BIN2 is reduced. Then, the dephosphorylated C3H15 protein accumulates in the nucleus, where C3H15 and BZR1/BES1 directly suppress each other. Furthermore, C3H15 antagonizes BZR1 and BES1 to regulate cell elongation by inhibiting the expression of their shared cell elongation-associated target gene, *SMALL AUXIN-UP RNA 15* (*SAUR15*). Importantly, rice CCCH protein LEAF AND TILLER ANGLE INCREASED CONTROLLER (OsLIC) was identified as another antagonistic transcription factor of BZR1 to attenuate BR signaling at high levels [45,46]. The gain-of-function mutant *lic-1* or *OsLIC* overexpression lines had erect leaves, a phenotype similar to that of BZR1-depleted lines.

Considering the importance of plant CCCH proteins in BR signaling, we investigated the responses of other 60 Arabidopsis *CCCH* genes to BR treatment using qRT−PCR assays. As illustrated in Figure 3, 2,4−eBL treatment resulted in a significant alteration in the expression levels of most of *CCCH* genes. Based on the different 2,4−eBL concentrations and the corresponding *CCCH* expression levels, three CCCH groups were generally classified. In group I, the expression levels of CCCH genes, including *AtC3H28*, *AtC3H30*, *AtC3H46*, *AtC3H55*, *AtC3H62*, *AtC3H63*, *AtC3H64* and *AtC3H65*, were activated at low concentrations (≤10 nM) of 2,4−eBL, while the expression of group II CCCH genes, such as *AtC3H42*, *AtC3H43*, *AtC3H44*, *AtC3H45* and *AtC3H53* was suppressed at high concentrations (≥100 nM) of BL. The third group contained genes *AtC3H1*, *AtC3H7*, *AtC3H16* and *AtC3H67*, which were induced at high concentration (≥100 nM) of BL but suppressed at low concentrations of BL.

### 3.5. SA Signal Response

SA plays a central role in plant innate immunity [54]. In Arabidopsis, AtC3H14 inhibits basal defense against *Pst* DC3000 by affecting SA level and signaling [27] (Figure 2). Challenge with *Pst* DC3000 or the flagellin peptide flg22 enhances the phosphorylation of C3H14 by MPK4 in the cytoplasm, attenuating the C3H14-activated expression of its targets *NIM1-INTERACTING1* (*NIMIN1*) and *NIMIN2*, two negative regulators of SAR. SA treatment reduced the MPK4 protein levels but induced the accumulation of dephosphorylated C3H14 protein, resulting in the higher expression of *NIMIN1/2* in Myc-C3H14OE plants, indicating that SA affects the action of MPK4-C3H14-NIMIN1/2 cascades in immunity. However, the constitutive activation of MPK4 in Myc-C3H14OE plants did not affect the SA-induced accumulation of phosphorylated C3H14 protein, suggesting that SA signaling mediated by the C3H14-NIMIN1/2 cascades may be independent of MPK4 phosphorylation.

AtCPSF30 is a member of the subunits in the polyadenylation complex, and functions as a general positive controller of the SA pathway [41]. The mutation of *AtCPSF30* led to the dramatic suppression of SA pathogenesis-related markers *PR1*, *PR4* and *PR5*. In addition to functioning in both the ABA and JA pathways, AtTZF2/AtC3H20/AtOZF1 promotes NONEXPRESSOR OFPR1(NPR1)-independent SA signaling and thereby enhances defense against bacterial pathogens [17]. In pepper, CaC3H14 was identified as a SA-related defense regulator [48]. The virus-induced silencing of *CaC3H14* increased susceptibility to *Ralstonia solanacearum,* accompanying a reduced expression level of SA-dependent *PR1,* while *CaC3H14* overexpression in tobacco triggered the expression of SA-dependent genes.

### 3.6. Auxin Signal Response

Auxin is a key modulator of plant architecture, including organ building, vascular cell division, cell elongation, apical advantage and root tropism [55]. Arabidopsis CCCH protein HIZ1 regulates auxin-induced root growth by interacting with HAKAI, a protein required for m^6^A methylation [43]. Transgenic plants overexpressing *HIZ1* driven by 35S promoter exhibited developmental defects, including the decreased level of m^6^A, shorter primary roots and reduced lateral root induction, similar to the phenotypes of the major hypomorphic m^6^A writer mutants in response to NPA and NAA treatment. AtCPSF30 is a member of the subunits of plant polyadenylation complex and its T-DNA insertion mutant *oxt6* showed reduced fertility, reduced lateral root number and altered sensitivities to auxin [42].

Poplar PaC3H17 is an ortholog of Arabidopsis AtC3H14 and 15, and it acts as a direct target of PaMYB3 and PaMYB21, both which are second-level master switches of the transcription network for wood formation [30]. Similar to the role of C3H14 and 15 in stem development, PaC3H17 positively regulates secondary cell wall thickening in the poplar stem. Subsequent study shows that PaC3H17 and its target PaMYB199 form an auxin−responsive functional complex, promoting cambium division by a dual regulatory mechanism [31] (Figure 2). After auxin treatment, *PaC3H17* transcription was enhanced, which attenuated the PaMYB199−driven suppression of cell division−associated target gene expression. The interaction ability of PaC3H17 with PaMYB199 was also increased, which further promoted the transcriptional repression of *PaMYB199* expression.

## 4. Prediction of the Responses of Arabidopsis CCCH Proteins to Several 

### Major Hormones

As we described above, Arabidopsis CCCH genes regulate a variety of biological processes through multiple hormone pathways. We here used online Arabidopsis eFP browser (http://bar.utoronto.ca/efp/cgi-bin/efpWeb.cgi accessed on 20 July 2022) to predict which CCCH genes are responsive to the six major hormones, including ABA, GA, BR, JA, ET and auxin. Thirty-six CCCH genes were selected due to the presence of their probes in the database. Except for *AtC3H12*, all 35 genes showed altered expression levels when treated with one of the six hormones (Figure 4). Among them, 16 genes were altered in response to at least three hormones. Some genes have already been experimentally validated. For instance, *AtTZF2*/*AtC3H20*/*AtOZF1* overexpression confers ABA hypersensitivity, enhances drought tolerance and delays JA-induced senescence [16]. AtTZF2 also positively regulates NPR1-independent SA signaling and defenses against bacterial pathogens [18]. AtC3H15 has been corroborated to mediate the BR, JA and ET signaling pathways to control growth, development and stress responses [24,25,26,27,28,29] (Figure 2). Importantly, three genes, *AtC3H5*, *AtC3H11* and *AtC3H28*, were predicted to react to all six hormones. More studies need to be performed to investigate how these CCCH proteins integrate multiple hormones to regulate the lifecycle.

The prediction data indicated that 12 CCCH genes, including *AtC3H20*, *AtC3H25*, *AtC3H28*, *AtC3H42*, *AtC3H43*, *AtC3H45*, *AtC3H46*, *AtC3H53*, *AtC3H56*, *AtC3H57*, *AtC3H62* and *AtC3H63*, were responsive to BR (Figure 4), which is consistent with the qRT–PCR result (Figure 3). It is worth noting that some CCCH genes, such as *AtC3H2*, *AtC3H3*, *AtC3H17*, *AtC3H36* and *AtC3H59*, showed different BR responses between the prediction and qRT−PCR results. It is possible that the inconsistency originated from different BL concentrations used in the treatments, different treating time and different plant developmental stages. For example, 2–week–old seedlings were treated with different concentrations of 2,4−eBL for 1 h in our qRT−PCR analysis, while the results of the 7–day–old seedling treated with 10 nM 2,4–eBL for 30 min were adopted in the Arabidopsis eFP browser.

## 5. Regulatory Mechanisms of CCCH Proteins in Plants and Animals

Most of animal CCCH proteins control a variety of cellular processes at the post-transcriptional level [56] (Figure 5). For example, the best known CCCH protein TTP often binds to class II AU-rich elements (AREs; AUUUA) in the 3′-untranslated region (UTR) of target genes including tumour necrosis factor-a (TNF-a) and promotes poly(A)-tail shortening by recruiting the CCR4−CAF1−CNOT1 complex in processing bodies (PBs) or stress granules (SGs) [56,57,58]. It is followed by mRNA degradation catalyzed by the 3′-5′ exoribonuclease exosome complex and the 5-cap decapping enzymes DCP1/DCP2 [10]. In this signaling cascade, *TTP* expression is activated by the released NF-κB in the nucleus derived from IκBα after phosphorylation and degradation in the cytoplasm (Figure 5). The CCCH protein Roquin 1 recognizes stem–loop motifs in the 3′ UTR of its target mRNAs and promotes mRNA decay by recruiting DDX6 and EDC4 [59,60], suggesting that animal CCCH proteins promote mRNA decay by different regulatory modes. hTTP is capable of activating the expression of genes that are expressed later in the response to insulin or serum, indicating that hTTP also functions as a transcription factor [61]. In addition, hTTP interacts with different types of proteins, such as protein kinases p38MAPK and MAPK kinase-2 (MK2), serine-threonine phosphatase PP2A, 5′ to 3′ exonucleolytic enzymes and CCR4−NOT deadenylase complex to perform post-translational modifications [57,58].

Plant CCCH proteins rely on protein−DNA, protein−RNA and protein−protein interactions to regulate growth, development and stress responses, similar to the regulatory mechanisms of animal CCCH proteins (Figure 5).

(1) At the transcriptional level. Arabidopsis AtC3H14 and 15 have the capacity of binding to single-strand DNA and double-strand DNA in vitro [24,25]. By using bioinformatics and EMSAs, the *cis* elements (CHRE, GGGAGA) that are recognized by AtC3H14 and 15 were identified in the promoters of target genes [29]. Genetic and biochemical data confirmed that both proteins regulate cell elongation, pollen development and basal defense against bacteria and fungi by specially binding to the CHREs in the promoters of different signaling components [25,26,27,29] (Figure 2). Transcriptional regulation of CCCH proteins was also found in other plant species. In rice, OsLIC functions as a negative component of BR signaling pathway and it can recognize the core element (CTCGC) in the promoters of *BZR1* and *OsIBH1* to suppress their expression [46]. OsSAW1 binds to a fragment (−1013 bp to −647 bp relative to the start codon) of the *OsGA20ox3* promoter to activate its expression level [40]. In sweet potato, *Ib*C3H18 directly regulates the expression of the ABA receptor gene *PYL8*, abiotic stress-responsive gene *SOS5* and ROS scavenging-related gene *CCS* [38].

(2) At the post-transcriptional level. Arabidopsis AtC3H14 and 15 inhibit cell elongation also by post-transcriptional regulation. Both proteins can bind to ribohomopolymers (Poly U, Poly C, Poly G) in addition to DNA in vitro [24,25]. By using RNA immunoprecipitation (RIP) and RNA electrophoretic mobility shift assays (RNA-EMSA), the ARE motif in 3′ UTR of two cell wall-associated genes *ADPG1* and *DUF1218* were shown to be recognized by AtC3H14 [24]. Arabidopsis KHZ1 and KHZ2 are also RNA-binding proteins. They repressed the splicing efficiency of FLC.1 by down-regulating the transcript level of *FLC* [33]. A study in rice indicated that OsTZF1 inhibited the expression of stress-responsive target genes by binding to the U-rich regions in the 3′UTR of their mRNAs [23]. Furthermore, the repression effect of OsTZF1 on these genes was enhanced by ABA, JA, SA or H_2_O_2_ treatment. Similar to OsTZF1, AtTZF1 is able to trigger the decay of ARE-containing mRNAs in vivo [62]. Future work should focus on how plant CCCH proteins regulate RNA decay, splicing, turnover and polymerization.

(3) At the protein level. Plant CCCH proteins interact with diverse proteins to regulate different development and stress responses. For instance, Cotton GhZFP1 enhances salt stress tolerance and fungal disease resistance in transgenic tobacco by interacting with GZIRD21A and GZIPR5 [47]. Arabidopsis HIZ1 interacts with HAKAI, forming a m^6^A writer complex to modulate auxin response and root development [43]. In sweet potato, the interaction between IbC3H18 and IbPR5 is strongly induced by NaCl, PEG6000, H_2_O_2_ and ABA [38]. We found that Arabidopsis AtC3H14 interacts with and is phosphorylated by MPK4 in the cytoplasm after flg22 induction, which decreases its transcriptional repression of immune−related target genes in the nucleus [27]. Likewise, AtC3H15 interacts with and is phosphorylated by BIN2, a negative component of the BR signaling pathway, leading to the inhibition of target gene expression and cell elongation [29]. Similar to AtC3H15, rice OsLIC inhibits target gene transcription and alters leaf and tiller angle by OsBIN2−mediated phosphorylation in BR signaling [46]. It is noted that the phosphorylation of AtC3H14/15 and OsLIC proteins by the corresponding partners inhibits transcriptional reprogramming in the nucleus, which is different from the effect of the TTP phosphorylation that results in a loss of ability to degrade target mRNA in the cytoplasm [63].

## 6. Future Perspectives

As illustrated in this review, plant CCCH proteins have different patterns of cell localization for distinct functions (Table 1). The majority of these CCCH proteins are localized in the nucleus as transcription factors, such as Arabidopsis SOMNUS [20], PEI1 [21] and AtC3H17 [64]. Some CCCH proteins can traffic between the nucleus and the cytoplasm, similar to animal CCCH proteins [63]. For example, Arabidopsis AtC3H15 interacts with and is phosphorylated by BIN2 in the cytoplasm in the absence of BR but it traffics into the nucleus and inhibits cell elongation after BR treatment [29]. Consistent with this, two putative leucine-rich nuclear export signal (NES) were identified in the N-terminal region of AtC3H15, which is responsible for the nuclear export of protein [25]. In fact, there is at least one NES in 54 members of Arabidopsis CCCH proteins and 62 members of poplar CCCH proteins [6], suggesting that these CCCHs may be nucleocy to plasmic shuttling proteins. Introducing fluorescence signal in functional studies may be an effective way to clarify the complex regulatory mechanisms of plant CCCH proteins in the cytoplasm and the nucleus.

Despite our current understanding of some hormones influencing the expression of *CCCH* genes having been substantially advanced in plants, it remains largely unclear how these CCCH proteins mediate the multifaceted regulation of hormone homeostasis and signal transduction. In particular, many CCCH−mediated regulatory events at the post-transcriptional level, such as RNA metabolism and protein interaction in hormone signaling still await to be uncovered. As described above, some Arabidopsis *CCCH* genes are strongly responsive to several major hormones (Figure 4). This raises the questions of how the CCCH protein-involved hierarchies and networks are established and modulated in response to a given hormone and how the CCCH proteins integrate multiple hormone signaling pathways. Moreover, it is necessary to dissect the hormone-mediated regulation of plant CCCH genes at higher spatiotemporal resolution. In this context, single-cell and tissue/cell−type−specific “omics” studies, including transcriptomics, metabolomics or (phospho) proteomics, will be the choice of approach in order to gain insights into the intricate cross-talks between plant CCCH proteins and phytohormones.

## Figures and Tables

**Figure 1 ijms-23-14288-f001:**
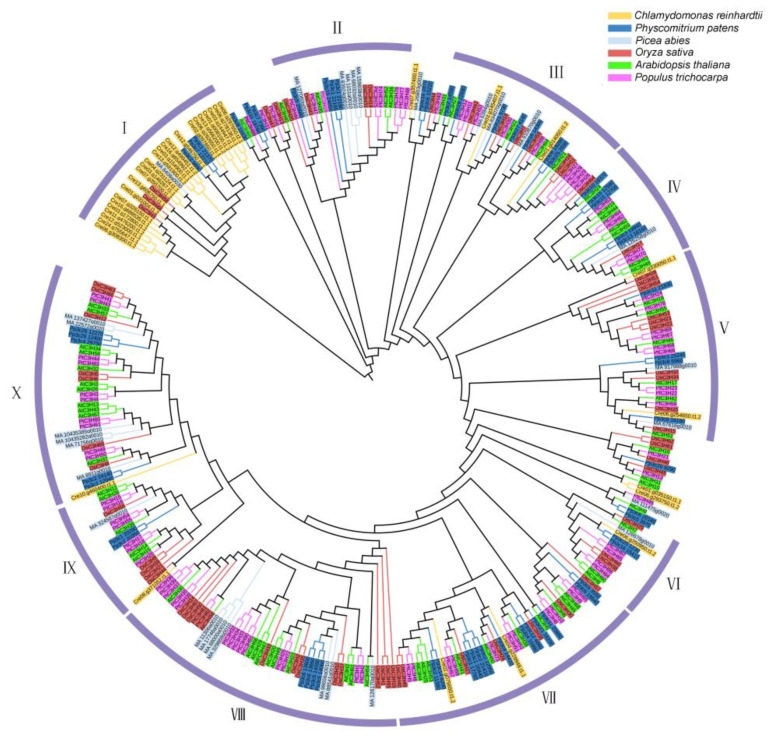
Phylogenetic analysis of CCCH proteins from six representative plants. The neighbor–joining (NJ) tree was constructed based on the alignment of full–length CCCH protein sequences from green algae (*Chlamydomonas reinhardtii*), moss (*Physcomitrium patens*), Arabidopsis (*Arabidopsis thaliana*), rice (*Oryza sativa*), poplar (*Populus trichocarpa*) and spruce (*Picea abies*). MEGA 6.0 was used with 1000 bootstrap replicates.

**Figure 2 ijms-23-14288-f002:**
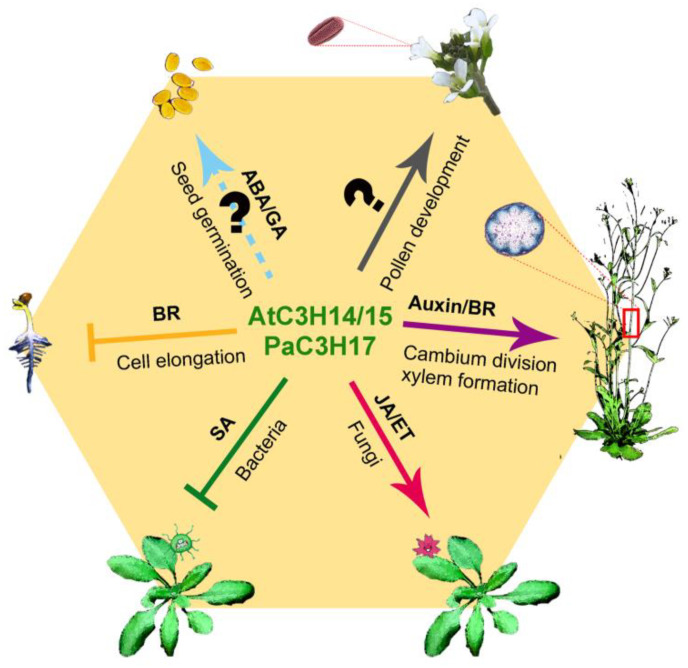
The functions of Arabidopsis AtC3H14, 15 and their poplar ortholog PaC3H17. These CCCH proteins regulate cell elongation in hypocotyls and stems, pollen development, xylem formation and basal defense against bacteria and fungi through the integration of multiple hormones, including BR, auxin, SA and JA/ET. The positive and negative regulations are represented by arrows and T-shaped lines, respectively. A dotted arrow shows unclear hormone regulation.

**Figure 3 ijms-23-14288-f003:**
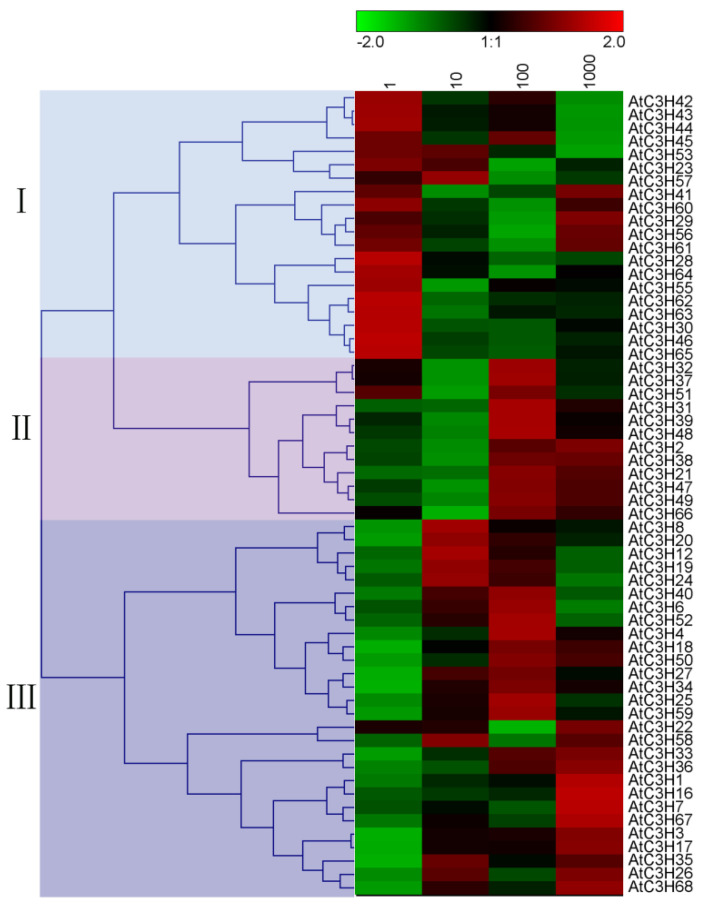
Relative expression levels of Arabidopsis CCCH genes in response to BR. Two−week-old wild−type Arabidopsis seedlings were treated with different concentrations (nM) of 2,4−eBL for 1 h and sampled for qRT−PCR analysis. Based on the effects of BR on gene expression, three groups (I, II and III) marked with different color backgrounds were classified. Data were normalized and hierarchically clustered based on Pearson correlation. Color scale at the top of heatmap represents log_2_ expression value.

**Figure 4 ijms-23-14288-f004:**
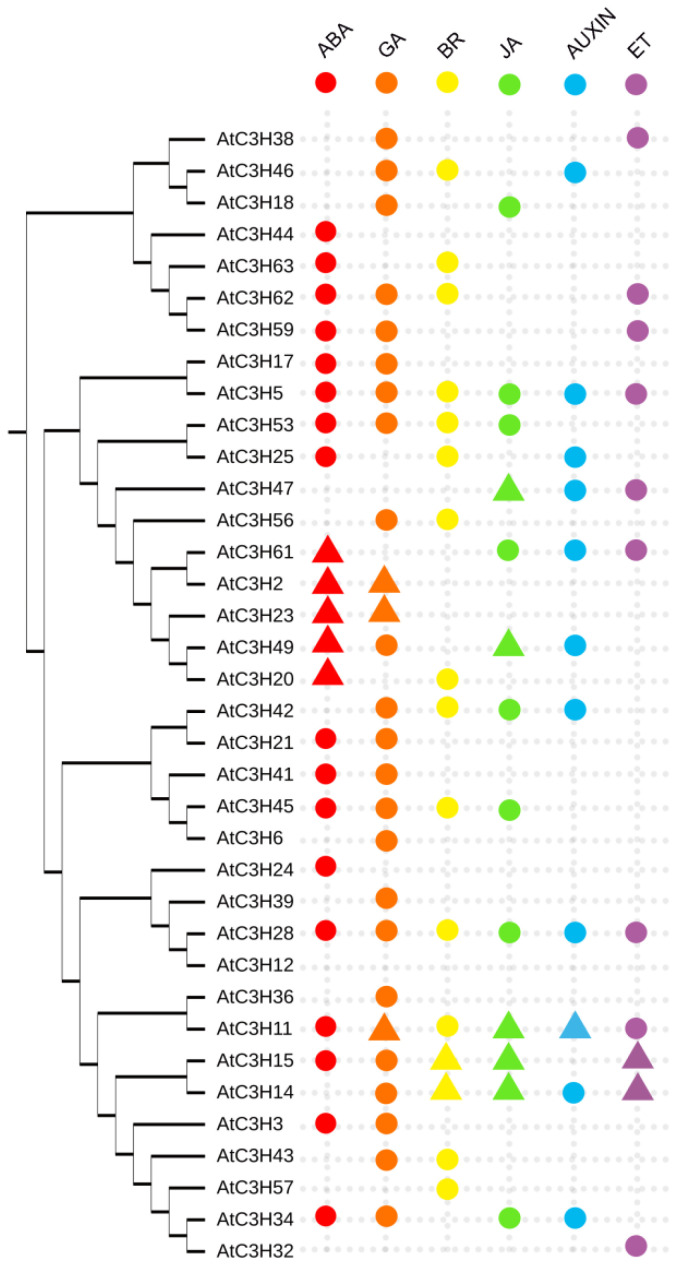
Responses of Arabidopsis CCCH genes to multiple hormones. The responses of 36 CCCH genes to six hormones (represented by different colored dots or triangles) were predicted using the Arabidopsis eFP browser. The phylogenetic tree was constructed based on the alignment of full–length Arabidopsis CCCH proteins. The CCCH protein shown with triangle has been characterized functionally.

**Figure 5 ijms-23-14288-f005:**
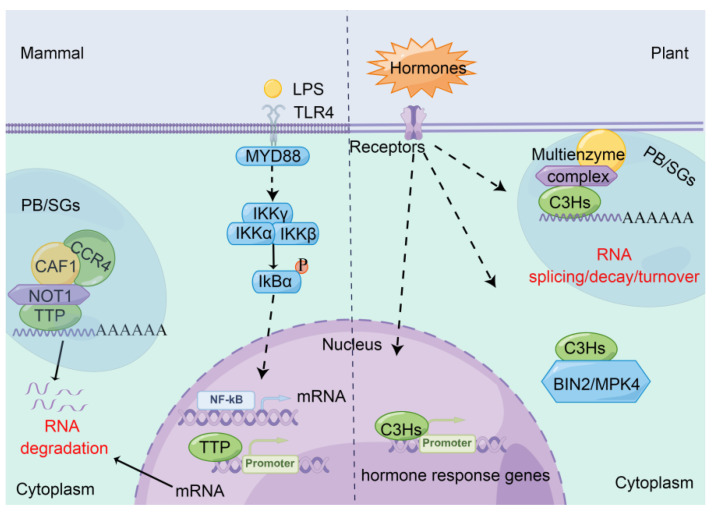
Regulatory mechanism models of plant and animal CCCH proteins involving in hormone signal transduction. Left panel: In mammals, lipopolysaccharide (LPS) recognizes Toll-like receptor 4 (TLR4), which activates the Nuclear Factor-κB (NF-κB) kinase complexes (IKKα, IKKβ and IKKγ), resulting in the phosphorylation and degradation of IkBa. The released NF-κB translocates into the nucleus and activates the expression of genes, such as tristetraprolin (TTP) and tumour necrosis factor (TNF). TTP further binds to AU-rich elements (AREs) in the 3′ UTR of TNF mRNAs and promotes mRNA decay by recruiting the CCR4−CAF1−NOT1 deadenylase complex in processing bodies (PBs) or stress granules (SGs). In addition, TTP performs transcriptional regulation. Right panel: In Arabidopsis, multiple hormones recognize the corresponding receptors, activating a serials of signal cascade reactions and inducing transcriptional regulation of AtC3H14 and 15 (the homologs of TTP) in nucleus. AtC3H14 and 15 not only interact with different types of proteins, such as MAP kinase 4 (MPK4) and BR-INSENSITIVE 2 (BIN2) in signal transduction, but also perform post-transcriptional regulation in PBs and SGs.

## Data Availability

Data is presented in manuscript.

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
