# Peer review of "Hormone Regulation of CCCH Zinc Finger Proteins in Plants"

_ijms, 2022, doi:10.3390/ijms232214288_

Round 1

Reviewer 1 Report

Comments to the Author

In the article entitled “Hormone regulation of CCCH zinc finger proteins in plants”, the authors attempt to review the hormone-governed biological functions in model plant species.

Heading 2

The author have performed a comparative phylogenetic study in representative species to identify evolutionary relationships.

1.     Except Group I, do authors noted any other phylogenetic separation to support their claim that CCCH proteins are conserved evolutionarily. The authors should elaborate this section further to find any evolutionary relationship of CCCH family of primitive plants [algae (green algae) and bryophyte (moss)] to higher vascular plants i.e. angiosperms [monocot (rice) and dicot (Arabidopsis)], and gymnosperm (poplar). Please support your finding with other studies of CCCH proteins in plants.

2.     How many groups CCCH proteins have in general? Please discuss other studies of CCCH classification in plants. How the ten groups classified in this comparative study is related to other studies. More importantly, can this classification be related to hormone response?

Heading 4

“Thirty-six CCCH genes were selected due to the presence of their probes in the database. Most of genes showed altered expression levels when treated with one hormone. Of them, half of genes were responsive to at least three hormones…”

1.     The statement is confusing, please mention the numbers or percentage of genes predicted to be affected by one hormone, at least three hormones, all six hormones.

2.     Do authors find experimental evidences to support this prediction? If so, please discuss in section 4

Heading 5

The similarity in the mechanism of animal hTTP and Arabidopsis C3H14 seems superficial in terms of LPS vs. hormone receptor, the promoter binding elements, and protein target portion. The authors should discuss the information available for plant CCCH and concisely compare the information with hTTP wherever applicable.

Reviewer 2 Report

In this review, authors have attempted to describe the role of CCCH zinc finger protein in mediating developmental and adaptive processes by hormonal regulation and vice versa. Authors have also created phylogenetic trees to show the evolutionary conservation and checked the expression of 60 CCCH ZFP in response to the BR hormone.

This is an important topic and needs to be pushed forward, and therefore I believe the review needs to be explicable. This review requires improvement and corrections.

1.    The introduction section does not provide adequate details of the CCCH ZFP. For example, the review does not describe TZF (Tandem CCCH Zinc Finger proteins). It does not describe RR-TZF. Similarly, the addition of the regulatory mechanism of CCCH will improve the introduction. For reference, the reviews by Jang 2016, and Pomeranz et al, 2011, have described the CCCH ZFP in detail. 

2.    The language and flow of the text make it difficult to understand the message authors want to convey. Some sections are just scattered collections of reports. Various reports are just touched upon and were not elaborated and thus making it difficult to follow. For example, how ORA59 gets modulated and how it affects immunity against B. cinerea. The authors just mentioned that ORA59 gets “modulated”, not clear if it was enhanced or decreased, or degraded.

3.    Similarly, throughout the review authors just used terms like modulation, regulation, affects, etc, however, do not mention whether regulation was positive or negative regulation, or whether expression was increased or decreased. e.g. p3l94, p3l96, p3l101 (change ABA content), and various other places.  

4. Correction needed on page p3l88, the author mentions “in contrast” however, it's not in contrast. By antagonizing ABI2 (negative regulator), AtTZF positively regulates ABA signaling. 

Round 2

Reviewer 1 Report

The authors have incorporated all the suggestions/observations ask during the previous review. I recommend the paper be accepted now.

Author Response

Thanks for your positive evaluation.

Reviewer 2 Report

The manuscript has been significantly improved and can be accepted after a few minor changes.

In figure 2, indicating whether the regulation is positive or negative using different arrows will provide better insight.

In line 134, the authors mentioned that AtTZF2/3 is involved in JA senescence. It is in contrast to the fact that it antagonizes the JA-induced senescence and delays the senescence. The line can be improved.

In line 202, the authors can mention the type of regulation.

In line 184/185, the addition of how SA signaling mediates C3H14/NIMIN1/NIMIN2 cascade independent of MPK4 phosphorylation will enrich the text.
